# ARNOLD: A Benchmark for Language-Grounded Task Learning with Continuous States in Realistic Scenes

**Ran Gong**[1*], **Yizhou Zhao**[1*], **Xiaofeng Gao**[1], **Jiangyong Huang**[2,4], **Qingyang Wu**[3],
**Wensi Ai**[1] , **Ziheng Zhou**[1], **Baoxiong Jia**[1,2], **Song-Chun Zhu**[2,4,5], **Siyuan Huang**[2]
[1]Center for Vision, Cognition, Learning and Autonomy, UCLA
[2]Beijing Institute for General Artificial Intelligence (BIGAI)
[3] Columbia University [4] Peking University [5] Tsinghua University

**Abstract:** Understanding continuous object states is essential for task planning since they are generally not discrete in the real world. However, most previous task learning benchmarks assume discrete (*e.g.*, binary) object states, making it barely applicable to transfer the policy from the simulated environment to the real world. Moreover, The trained robot is limited to follow human instructions by grounding actions and states. To address such challenges, we present ARNOLD, a benchmark that evaluates language-grounded task learning with continuous states in realistic 3D scenes. ARNOLD consists of 8 language-conditioned tasks that involve understanding object states and learning policies for continuous goals. To encourage language-instructed learning, we provide template-generated demonstrations with language descriptions. We will benchmark the task performances with state-of-the-art language-conditioned policy learning algorithms. We will release ARNOLD and host challenges to promote future research in embodied AI and robotics.

**Keywords:** Grounded task learning, Continuous states, Simulated environment

## 1 Introduction

One key capability that emerged from the long evolution of human beings is the ability to ground language. Such capability enables humans to describe and learn concepts, execute tasks, and also communicate with others. With the recent culmination of grounding concepts in images [Radford et al., 2021; Kamath et al., 2021; Saharia et al., 2022], few have studied the grounding of actions [Shridhar et al., 2022; Zheng et al., 2022]. Considering how humans understand object status and relate language instructions to the physical world, a natural question to ask is: How can we give robotics systems the same capability to understand and execute language instructions in the physical world?

There exist several challenges that cause non-trivial difficulties in the learning of robotic systems. First, robot tasks depend highly on detailed scene information such as geometry information, layouts, and visual appearances[Xing et al., 2021]. This information requires robots to acquire detailed scene understanding by extracting useful geometrical features for task execution. This challenge is further aggravated by the various combinations of different scene configurations, including novel appearances, objects, and spatial positions. Therefore, it is critical for robotic systems to learn and generalize skills of a task to novel scene configurations.

Moreover, an essential capability of humans lies in the precise understanding of desired goal states. Although humans often refer to goals with simple descriptions (*e.g.*, a cup *half* filled, a door *fully* opened, *etc.*), what we understand is actually the status of physical properties (*e.g.*, half the volume, pulled to 180°, *etc.*). Given this abstraction, it is exceedingly difficult for robots to learn the accurate goal state from abstracted task descriptions, not to mention more abstract language descriptions that refer to an implicit range of continuous object states (*e.g.*, *a bit of* coffee, *slightly* open, *etc.*). This

6th Conference on Robot Learning (CoRL 2022), Auckland, New Zealand.

further requires the understanding of continuous object states. To consider such subtle capabilities, robot systems need to maintain a mapping from language instruction to an estimate of desired goal states before task execution.

A first step towards the aforementioned robot learning problems is to build robot simulation systems that facilitate language-grounded learning. In fact, recent years have witnessed significant progress in simulated environments that facilitate grounded task learning [Das et al., 2018; Shridhar et al., 2020; Mees et al., 2021; Zheng et al., 2022]. Despite the impressive performance of these benchmarks, they suffer from several limitations that hinder the robots' ability to operate in real-world environments effectively: 1) assuming discrete (*e.g.*, binary) object states and perfect motor control, therefore ignoring, low-level geometry and dynamics of the object [Szot et al., 2021; Srivastava et al., 2022; Ehsani et al., 2021], not requiring in-depth physical state understanding or fine-grained manipulation skills; 2) not grounding instructions to precise states [Zheng et al., 2022; Shridhar et al., 2022], omitting the challenging grounding problem from language to specific states in a continuous spectrum; 3) performing tasks in neat environments instead of in scenes spatially occupied by various surrounding objects and visually disturbed by diverse textured backgrounds [Kumar and Todorov, 2015; Lin et al., 2020; Zheng et al., 2022].

To better address the grounded task learning problem in a realistic setting, we introduce, ARNOLD, a new benchmark for grounding task description languages to **continuous robot actions** and **continuous object states** in a **photo-realistic** and **physical-realistic** interactive environment. ARNOLD is built on top of Nvidia Issac Sim, which provides accurate physics simulation and state-of-the-art rendering. We use a hybrid human-template-based approach to synthesize data for eight different tasks. Each task features continuous robot motion with friction-based grasping and object state manipulations. These tasks require different motor skills, including but not limited to grasping, pushing, pulling, and pouring liquid into a container for 40 unique objects in 20 different scenes. Then, we pair each demonstration with a template-based language instruction that describes task goals.

Unlike prior work, ARNOLD tests the agent's ability to generalize to unseen object states through understanding language instructions and continuous object states.

In summary, our grounded continuous task learning benchmark ARNOLD has the following contributions:

- A set of eight different **language-conditioned manipulation** tasks featuring different motor skills and diverse object states.
- A **photo-realistic** and **physical-realistic** 3D simulation environment with **continuous states** for different objects and fluids.

## 2   Related Work

**Simulation Environment for Embodied AI.** There has been a large number of works that simulate indoor household activities for training and evaluating AI agents [Kolve et al., 2017; Puig et al., 2018; Li et al., 2021; Szot et al., 2021]. Most of these simulators only simulate preconditions and post-effects of agent actions or use simplified state representations. Instead, our work simulates continuous states for articulated objects and fluids at the particle level.

Some works use simplified abstract discrete action space. Abstract discrete action space can reduce the task's difficulty. However, agents trained in this setting are unaware of the low-level geometry and dynamics of the objects, which would restrict their possibility of transferring to the real world. For example, grasping is often simplified by attaching a nearby object to the gripper [Ehsani et al., 2021; Shridhar et al., 2020; Srivastava et al., 2022; Szot et al., 2021], or through contact [Li et al.; James et al., 2020]. In contrast, we control robots with 7-DOF continuous control and friction-based grasping powered by a state-of-the-art physics engine (PhysX5.0).

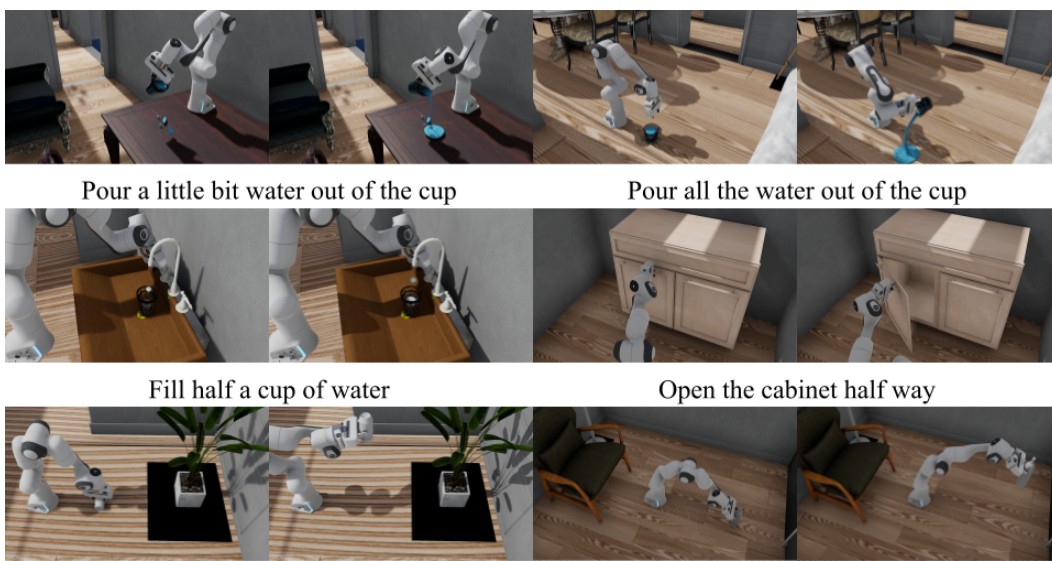

Figure 1: Examples of tasks in ARNOLD. For accomplishing these tasks, it requires visual recognition, text understanding, as well as diverse cognitive skills.

Among works that simulate continuous object state change, Gao et al. [2019] predict an action verb giving task goals. However, they do not manipulate object states in a fine-grained manner and only assume binary object states. Softgym[Lin et al., 2020] is an object manipulation benchmark that provides a realistic simulation of deformable objects. However, there is a lack of diversity in terms of objects and scenes. In contrast, ARNOLD provides a wide variety of scenes and objects. In addition, as Isaac Sim is a part of Omniverse, authoring more complex assets can be easily managed through other applications within the omniverse universe.

**Language Conditioned Manipulation.** Relating human language to robot actions has been of interest in recent research [Mees et al., 2021; Lynch and Sermanet, 2020; Shridhar et al., 2020; Zheng et al., 2022]. However, the environments in these works either lack realistic physics or do not have realistic scenes where the agent's surroundings will constrain its motion, and different scene objects might occlude the agent's viewpoint. Most importantly, prior work aims to ground human languages to static object properties, such as colors and shapes. By contrast, ARNOLD provides instructions for continuous object states.

**Continuous state understanding.** Some recent research tries to predict object states [Liu et al., 2017; Nagarajan and Grauman, 2018]. However, the object states are discrete rather than continuous. More recently, [Weng et al., 2021] tried to predict object states in a continuous spectrum. However, they do not involve manipulating objects from an arbitrary starting state to the desired state, and it only involves articulated objects. In addition, they do not model the language grounding process.

## 3  Arnold Benchmark

The aim of ARNOLD is to evaluate the learning of language-conditioned continuous control policy over a diverse range of physical-realistic and photo-realistic 3D scenes. In this setting, an agent needs to have the following capabilities:

1. Understand the goal state of the current object from human language instruction.

2. Estimate current object states from multiple camera inputs.

3. Propose and actuate motion plans based on physics over long-horizon.

| Benchmark | Alfred | Maniskill | Calvin | Behavior | KitchenShift | Vlmbench | **Ours** |
|---|---|---|---|---|---|---|---|
| Language | ✓ | ✗ | ✓ | ✗ | ✗ | ✓ | ✓ |
| Multi Camera | ✗ | ✓ | ✓ | ✗ | ✗ | ✓ | ✓ |
| Fluid | ✗ | ✓ | ✗ | ✓ | ✗ | ✗ | ✓ |
| Physics | ✗ | ✓ | ✓ | ✗ | ✓ | ✓* | ✓ |
| Continuous | ✗ | ✓ | ✓ | ✓ | ✓ | ✓ | ✓ |
| Scenes | ✓ | ✗ | ✓ | ✓ | ✗ | ✗ | ✓ |
| Robot | ✗ | ✓ | ✓ | ✓ | ✓ | ✓ | ✓ |

Table 1: **Comparison with other benchmarks.** ARNOLD features continuous control with **robots** over continuous object states with a large number of human demonstrations in photo-realistic scenes. Each task in ARNOLD is specified by a human language instruction. ARNOLD also leverages advanced physics simulations powered by PHYSX5.0 to simulate articulated bodies and fluids. **Language:** task goal is specified by a human language instruction. **Multi-Camera:** robot is equipped with multiple cameras. **Fluid:** advanced fluid simulation. **Physics:** realistic physics simulation. * RLbench-based benchmarks use simplified grasping. **Continuous:** object state is continuous. **Scene:** tasks are performed with scene background. **Robot** : perform actions with real robots.

Concretely, for an instruction like pouring half of the water out of the cup, the agent needs to understand the instruction and know what the cup should look like when it's half full from multi-model sensors and reasoning. Then it needs to further reason over the best sequence of actions to grasp the cup and pour the water out. Finally, it needs to know when and how to stop to meet the goal object state it determined previously.

## 3.1 Simulation Environment

### 3.1.1 Environment Components

**Simulation Platform.** ARNOLD is based on NVIDIA Isaac Sim[Makoviychuk et al., 2021], a robotics simulation application that enables the creation of photo-realistic and physically-accurate virtual environments to develop robots. The physics simulation in ARNOLD is based on PhysX 5.0, and the photo-realistic rendering is powered by GPU-enabled ray tracing. See Figure 1 for some examples of simulation and rendering. As demonstrated by Li et al., the rendering effect provided by Isaac Sim is much more realistic than any existing simulator.

**Scenes and Robots.** The scenes in ARNOLD are based on Fu et al. [2021], a large-scale synthetic indoor scenes dataset. As a result, ARNOLD scenes have professionally designed layouts and are populated by high-quality 3D models. Each scene contains a 7-DOF Franka Emika Panda manipulator with a parallel gripper.

**Fluids.** The fluids in ARNOLD are simulated using GPU-accelerated position-based-dynamics(PBD) method Macklin and Müller [2013] through omniverse. Then we perform surface construction through marching cubes Lorensen and Cline [1987] to obtain the final fluid rendering effect. For faster rendering, we make the surface construction step optional.

**Articulated Objects.** In addition to objects provided by Isaac Sim, we also get object parts from open-sourced dataset Kolve et al. [2017]; Xiang et al. [2020]. We perform modifications to a few object meshes, e.g. changing materials and adding covering meshes to cabinets and drawers, for a more natural appearance. We also perform convex decomposition to create realistic collisions for each object. Moreover, to ensure physically-realistic simulation, we assign physics parameters to objects, including weight and friction for rigid-body objects, cohesion, surface tension, and viscosity for fluids. We chose these parameters based on Mu et al. [2021] and human operator feedback.

### 3.1.2 Observation and Action Space of the Robot

**Action space.** The agent can control the 7 joints and the gripper of the manipulator. In addition, the agent can control the robot end-effector through the provided motion planner. Since the focus of this work is object manipulation, we do not allow the robot to navigate the room.

**Observation space.** The robot has five different camera views around it. One is on top of the robot, one is on the opposite side of the robot, one is on the left of the robot, and two are on the robot gripper. This setting is to avoid view occlusion problems during manipulation. Each camera provides RGB-D input. By default, each camera has a resolution of $128 \times 128$. Users can render arbitrary resolutions through the replay. In addition, we provide robot joint positions and velocities. To support learning, we provide additional state information available to access if needed: robot base position, object positions, object rotations, and object part semantic mask.

## 3.2 Task Design

### 3.2.1 Task Definition

In ARNOLD, each task $t \in T$ is defined by a tuple $(t_o \in O, t_a \in A_o, t_b \in B_s, t_g \in G_s)$. Here $O$ is the set of object types the robot needs to manipulate (e.g. cabinet, drawer, glass). $A_o$ is the set of attributes we care for object $o$ (e.g. joint angle of the cabinet, the water level in the container). $B_a$ is the set of initial attribute values of the object being manipulated, e.g. the initial joint angle of the cabinet or the amount of water in a cup at the beginning. $G_s$ is the set of required goal values of the object.

| Task Type | Goal variations | success condition |
|---|---|---|
| Pick up object | 10, 20, 30, 40 cm above | $\pm$ 5 cm |
| Reorient object | 45, 135, 180 degrees | $\pm$ 15 degree |
| Open cabinet | 25, 50, 75, 100 % open | $\pm$ 10 % |
| Open drawer | 25, 50, 75, 100 % open | $\pm$ 10 % |
| Close cabinet | 25, 50, 75, 100 % open | $\pm$ 10 % |
| Close drawer | 25, 50, 75, 100 % open | $\pm$ 10 % |
| Pour water | 25, 50, 75, 100 % of water | $\pm$ 10 % |
| TransferWater | 20, 40, 60, 80 % of water | $\pm$ 10 % |

Table 2: Task Description for eight different tasks. Each task features four different goal state variations specified by human language. For each task, the success signal will be triggered if the object state is within the success tolerance for two seconds. The tolerance threshold is developed based on a pilot study where we ask humans to use a gamepad to finish the task.

### 3.2.2 Task Types

In ARNOLD, there are nine types of tasks. Here we define each task type.

**PickUpObject.** $T_{pick} = \{t|t_o = bottle \wedge t_a = height \wedge (t_b < t_g)\}$. This task requires the agent to find the object, pick it up and raise it to the desired height. This task is meaningful in real-life human-robot collaboration where the human instructs the robot to raise the object to a certain height so he can receive the object from the robot.

**ReorientObject.** $T_{reorient} = \{t|t_o = bottle \wedge t_a = orientation\}$. To execute this task, the agent must find the object, pick it up, and reorient it to the desired angle. A real-life example of this task can be getting ketchup out of the glass bottle, which requires the agent to reorient the bottle to the desired angle.

**OpenCabinet/Drawer.** $T_{open} = \{t|t_o = \{Cabinet, Drawer\} \wedge t_a = jointVal \wedge (t_b < t_g)\}$. For this task, the agent needs to estimate the current joint position of the cabinet/drawer and take appropriate actions based on human language. For cabinets, the motions are constrained by a revolute joint. For drawers, the motions are constrained by a prismatic joint.

**CloseCabinet/Drawer.** $T_{open} = \{t|t_o = \{Cabinet, Drawer\} \wedge t_a = jointVal \wedge (t_b > t_g)\}$. This task is the reverse of the previous task. It is relatively easy compared to other tasks; however, it requires the agent to anticipate the consequences of its actions correctly. If the agent exerts a large force to push the object initially, the drawer will be closed completely.

**PourWater**. $T_{pour} = \{t|t_o = \{Container\} \wedge t_a = waterLevel \wedge (t_b > t_g)\}$. To successfully execute this task, the agent needs to pick up the cup and pour a desired amount of water out of the cup. Therefore, the agent needs to estimate the amount of water in the cup and take appropriate actions.

**TransferWater**. $T_{transfer} = \{t|t_o = \{Container\} \wedge t_a = waterLevel \wedge (t_b < t_g)\}$. The agent needs to pick up the cup and transfer a specified amount of water from one cup to another for this task. This task is the most difficult task of all. First, the agent needs to know where to pour water without knocking over the container cup. Then the agent needs to estimate how much water is in the cup and stop at the right time. To avoid ambiguity, the two cups are the same.

Detailed task descriptions are in table 2

### 3.2.3 Evaluation Metrics

**Task Success**. Each task is parametrized by a tuple as indicated in section 3.2.1. Task success is defined as in table 2. The goal state is satisfied when the object stays in a tolerance threshold, as defined below:

$$|t_a(i) - t_g| < \epsilon_t, \tag{1}$$

where $\epsilon_t$ is the success tolerance for each task, as shown in Table 2.

If the object state satisfies the goal state for 2 seconds, then we declare the task as a success. For example, consider the task: "pour half a cup of water out of the cup". The agent succeeds if, there are $40\% - 60\%$ of water particles in the cup for two seconds within the episode limit and the cup is back to upright.

### 3.2.4 Task split

To allow generalization on novel scenes and objects, we split the dataset into four folds: 1) training, 2) novel scene textures, where objects are seen during training but scene textures are different, 3) novel objects, where scenes textures are seen during training but not objects, and 4) novel language states.

## 4 Dataset Collection

### 4.1 Mission Definition

We call any particular robot task instance a mission. Each mission is defined by a tuple $(t \in T, i \in I_o, e \in E)$, which specifies the setting combinations of task $T$, object $I_o$, and environment $E$. So in a mission, the robot needs to perform a task $t$ in a specific environment $e$ using an object instance $i$. Moreover, we still allow the position of the robot and object in the scene to vary in a mission to create more diversity.

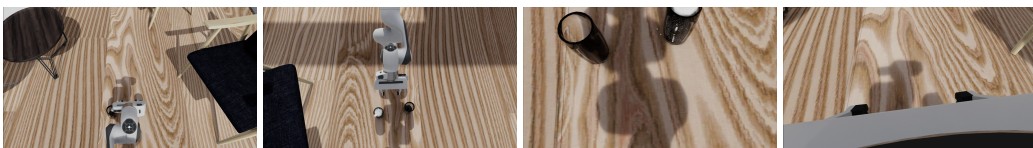
Figure 2: Multi View Camera Example

### 4.2 Human Demonstrations and Instructions

To understand how humans complete the task, we collect 2400 human demonstrations with 7 operators on all task types.

We created 400 missions $(t, i, e)$ consisting of various combinations of object instances, goal state values, and scenes. For each mission, the annotator would further specify two relative positions

| Task Name | Template | Example |
|---|---|---|
| PickObject | <v><o> | raise the bottle 20cm above the ground |
| ReorientObject | <v><o> | rotate the bottle 90 degrees from the up axis |
| OpenCabinet | <v><position><o> | pull the left cabinet half closed |
| CloseCabinet | <v><position><o> | close the top cabinet completely |
| OpenDrawer | <v><position><o> | open the bottom drawer midway |
| CloseDrawer | <v><position><o> | push the bottom drawer entirely closed |
| PourWater | <v> water <preposition><o> | pour all the water out of the cup |
| TransferWater | <v> water <preposition><o> | transfer almost half of the water to the glass |

Table 3: language generation template. We sample each attribute based during the generation process based on the proposed template. For each trajectory, we sample different language descriptions. (Where, <v>, <o>, and  stand for *verb*, *object*, and *state* respectively.)

between the robot and the object. For each position setting, the annotator would record two human demonstrations. So for each mission, we will record three human demonstrations.

We enable operators to teleoperate with the robot via the Xbox controller during human recording. To avoid view occlusion with other objects and furniture in the scene, we also allow operators to change the camera viewpoint using the controller joystick.

In addition, for each human demonstration collected, we sample 3 template-based language commands based on the language generation engine to describe the target object state. Note that the initial states are not specified in the command, thus requiring the agent to understand the current object state from observations.

Human demonstrations are served as a sanity check to ensure that all tasks can be completed. However, human demonstrations are very noisy. It poses too many difficulties for current models to extract meaningful information. To this end, we proposed a keypoint-based task template inspired by [Zheng et al., 2022; Shridhar et al., 2022]

### 4.3 Keypoint Based Task Templates

We manually mark keypoints around objects as in [Zheng et al., 2022] to generate noise-free trajectories. Each task composes of several stages. At each stage, a motion planner drives the robot toward a keypoint. When the stage condition is satisfied, the task planner will move on to the next stage. We designed a set of keypoint-based task planners for all objects and tasks in our benchmark. To generate feasible locations in the scene for objects and robots, we reuse human-annotated (object, robot) location pairs.

### 4.4 Instructions Generation

For each trajectory collected, we sample a template-based language command based on the language generation engine to describe the object state as described in Table 3. For example, "pour 50% of water out of the cup." Note that the initial states are not specified in the command, thus requiring the agent to understand the current object state from observations.

### 4.5 Dataset Statistics

In ARNOLD, We sample actions from an Xbox robot controller at 120 Hz resulting in 2488 trajectories or around 4,400,000 frames of image and action pairs. The recorded trajectory length is about 10.2 hours, with a median trajectory length of 13 seconds, an average of 14.8 seconds, a maximum of 91.6 seconds, and a minimum of 3.2 seconds.

Then for each (human, object) pair, we generate a trajectory through templates. Note, not all scenarios that can be solvable by humans are solvable by motion planning.

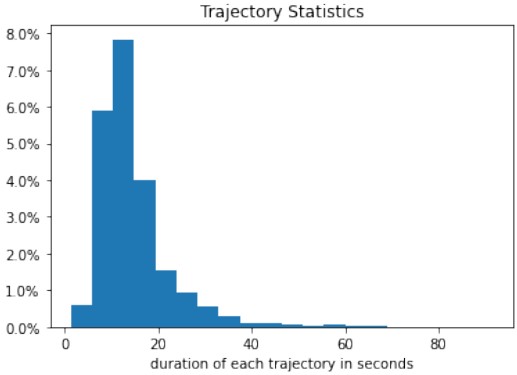

Figure 3: Trajectory Length Distributions

# 5 Baseline Model

Recently, there are two proposed baselines for language conditioned robot manipulation tasks.

6D CLIPort [Zheng et al., 2022] assumes a known task template and uses a top-down projection of fused point cloud to predict keypoint on the images space, then translates the projection back to the world space. To handle the missing height and roll from the top-down projection, they directly use a regressor to predict those two quantities. PerAct [Shridhar et al., 2022], instead, directly uses a voxel representaton to predict keypoints for different stages of the action.

The above two methods assume a strong spatial prior, making the prediction data-efficient and robust.

# 6 Limitations and Future Work

Even though we have put realism of the simulation as one of our key focuses, there is still some gap to the real world due to the precision limit of physical simulation and rendering. Rudin et al. [2022] has shown that sim-to-real is possible with policies trained under Isaac Sim. We plan to perform sim-to-real experiments in future work.

Some of our task settings are not very natural as we would see in the real world, though, due to limited 3D assets and limited manpower. For example, in real-life, people pour water into the sink instead of on the ground or the table except for cleaning purposes. But such a setting should not limit the robots from learning the critical manipulation tasks, which is the focus of this work.

Besides, robots are limited in their type variation. We collect data and perform the experiment on only one type of robot. There is no variation in the robot arm length and degree of freedom. We thought changing robot types might pose too much difficulty for our tasks.

Regarding the language instruction design, we did not use natural language generated by human annotators in this work. Template-based language is not as diverse as natural language. Therefore, we cannot achieve good coverage for human language descriptions to object states. At the same time, as indicated by Jansen [2020], language instructions generated by annotators, contain noises. So our instructions will also lack some real-world noise.

Besides, we allow operators to perturb the robot's relative positions against the object to avoid the overfitting indicated in Mees et al. [2021]. Operator-generated positions only cover a limited percentage of robot possible locations. More general settings should enable the robot to navigate. But this work focuses on object manipulation instead of navigation, so we put aside this complexity. We will consider adding it to our future work.

# 7 Conclusion

We presented ARNOLD, a new benchmark for language-conditioned continuous control over physical-realistic and photo-realistic scenes. ARNOLD contains eight different tasks requiring diverse skills. We further supply human demonstrations as well as template-generated demonstrations. ARNOLD tries to bridge the gap between object state understanding and object manipulation for modern robotics systems.

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

# A  Appendix for ARNOLD: A Benchmark for Language-Grounded Task Learning with Continuous States in Realistic Scenes

## A.1  Parsing Assets into Universal Scene Description (USD) format

Parsing assets into Omniverse is parsing graphic data into USD files. The goal is to make it easy for developers to read their assets (such as scene files, articulation bodies, animations e.t.c.) and have that information available when needed. This means that users can use a USD file with a bunch of different assets and then tell OMNIVERSE which ones are important for your game at any given time without having to go through each one individually.

### A.1.1  Working with 3D-Front dataset

The 3D-Front dataset is originally a collection of synthetic indoor scenes highlighted with professional design and a large number of rooms with high-quality textured 3D models. The source dataset contains three main parts: models, scene files, and textures. The dataset consists of more than tens of thousands of room layouts with thousands of furnished objects.

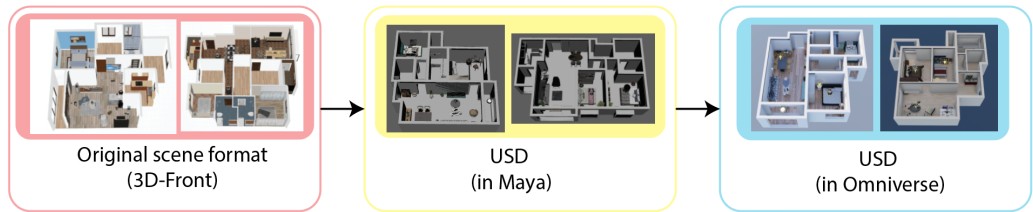

Figure 4: Pipeline for parsing scenes. First, we pre-process the original 3D-Front scene data. Then, we build an autommatic pipeline to load the scene layout into Autodesk Maya with custom designs. Finally, we convert the layout file into USD format and deploy it in Omniverse.

To parse the 3D-Front dataset into usd format, we apply the following steps:

- Parsing the original scene files (.json) into a data frame containing the mesh and furniture information;
- Using Maya MEL script to load scenes into Autodesk Maya;
- Applying Maya and Omniverse connverter to save the scenes into USD format.

### A.1.2  Working with Articulation bodies

The articulation bodies in our application mainly come from the SAPIEN. To parse the original ariculation bodies (.urdf) into USD format, we apply OMNIVERSE ISAAC SIM. The build-in tool within it allows to transfer urdf files into desired format.

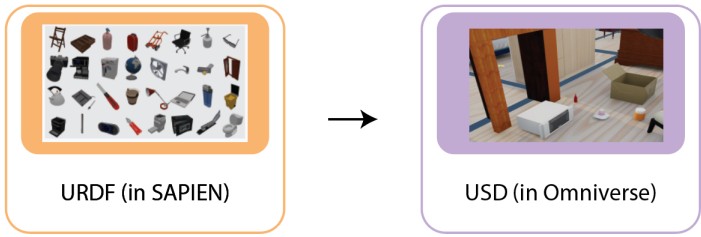

Figure 5: Pipeline for parsing articulation bodies. To convert the original articulation bodies in URDF format, we refine the built-in functionalities in Omniverse Isaac-Sim and partly modify the assets manually. Then we save the parsed objects in to USD format.

## A.2 Working with robotics

Robotics is a science that deals with the design, construction and operation of robots. The field encompasses many different disciplines such as computer science, electrical engineering, mechanical engineering and others. In Omniverse, robotics is built-in with sufficient details. Users may make it stronger by customizing the designs or even making their own robots from scratch.

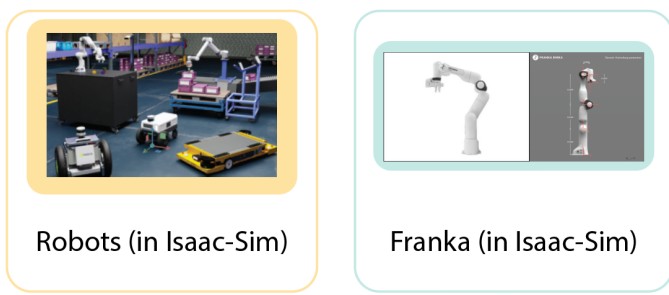

Robots (in Isaac-Sim)    Franka (in Isaac-Sim)

Figure 6: Robots in Omniverse Isaac Sim. The Omniverse platform provides a lot of features to simulating virtual robotics. It helpes researchers with the tools they need to build robust, physical-realistic simulations.

## A.3 Data Collection

### A.3.1 Robot Teleoperation

**Frame Transformation.** For data collection, we enable the operator to control the robot and the camera using an Xbox controller. We assume that the controller input is given in the robot base frame. As displayed in Figure 7, the base frame is define as the frame where the X axis is originated from the robot's base and pointing to the object and the Y axis is the up axis. The controller input is transformed to the world frame at each time step.

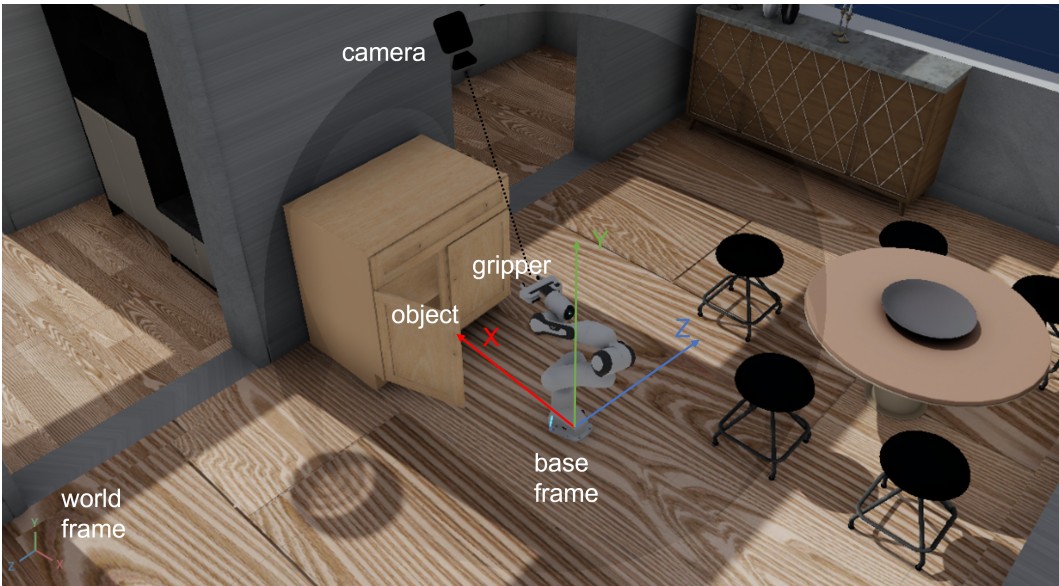

Figure 7: Frames and camera

**Teleoperation.** For robot teleoperation, the operator can change the position and rotation of the end effector and toggle the gripper (Figure 8). In addition to controlling the height and horizontal movement of the end effector, operators can change its rotation via two control modes (i.e. joint position control and end effector rotation control). The joint position control mode is intuitive since it

allows the operator to directly change the positions of joints A1 (shoulder joint), A6 (forearm joint) and A7 (wrist joint). The end effector rotation control allows operators to rotate the end effector around the X, Y, Z axis of the base frame while maintaining its position, and thus is a more general way of changing the rotation. The operator can switch between these two modes during the data collection.

**Camera control.** We enable operators to move their viewing camera freely in the virtual environment to avoid occlusions in cluttered scenes so that they can continuously monitor the end effector and the object. This is done by a spherical camera control (Figure 7), where the camera is continuously facing the end effector and the operator can move it around a sphere centered at the end effector. The radius of the sphere can also be adjusted for a more clear view.

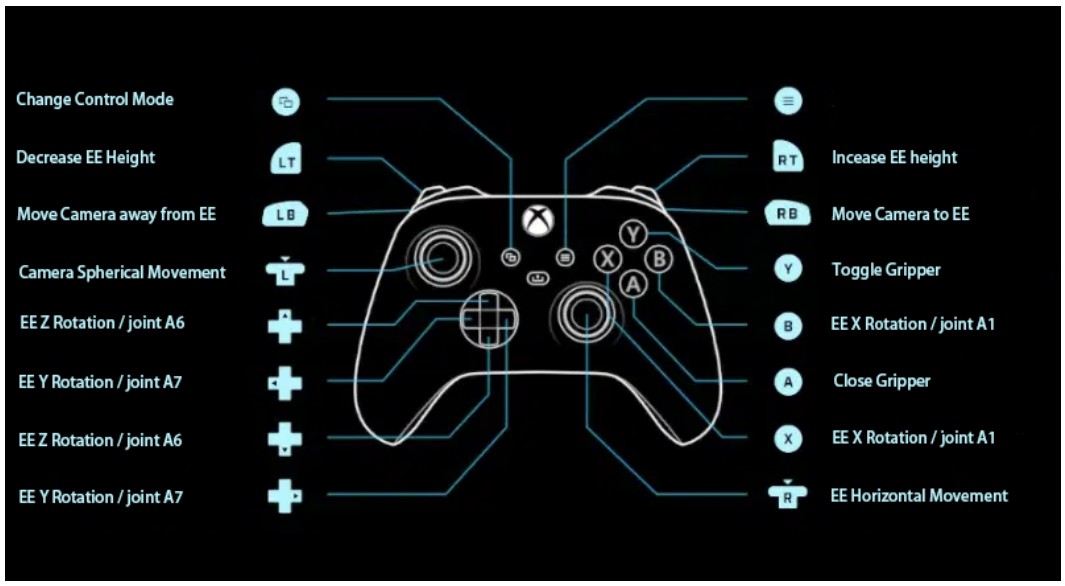

Figure 8: Controller Mapping

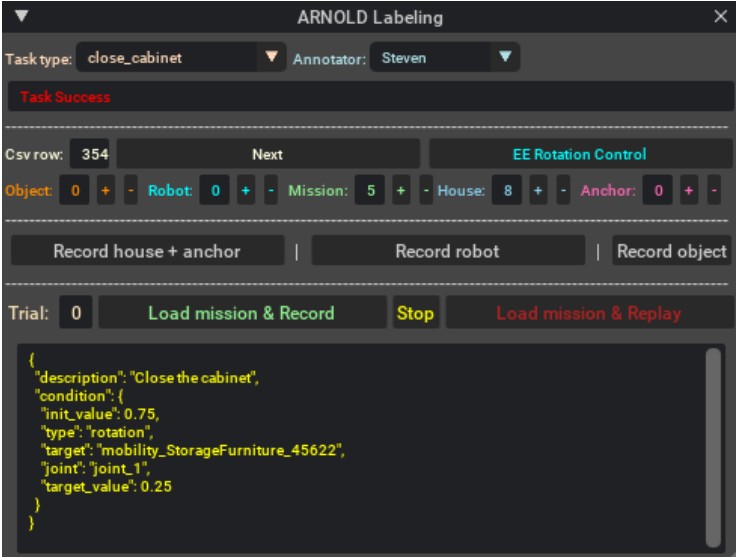

Figure 9: User Interface.

### A.3.2 Procedure

The human data collection process is done using a user interface (UI) Figure 9, which is implemented as an extension of Isaac Sim. For each mission $(t, i, e)$, the annotator is given a uniquely sampled tuple specifying the object instance, goal state values and scene. During the process, the annotator also needs to specify the transformation of the robot and object relative to the scene.

The annotator starts by moving the object group containing the robot and the object instance to an appropriate anchor place in the scene and clicking on the "Record house + anchor" button for recording. The annotator can then adjust the relative transformation between the object and the robot and click on the "Record Robot" button. For each mission, the annotator is required to change the transformation twice. After clicking on the "load mission and record" button, the annotator can control the robot to manipulate the object. In each trial, the goal is to change the attribute $t_a$ of object $t_o$ instance $i$ from $t_b$ to $t_g$. Two trials are needed for each mission with the adjusted transformation. Once a mission's success condition is met, the UI displays a "Task Success" message and the annotator can click the "Stop" button to stop the recording. Finally, the annotator can replay the recorded trajectory by clicking the "Load mission & Replay" button. To proceed to the next mission, the annotator can click on the "Next" button.

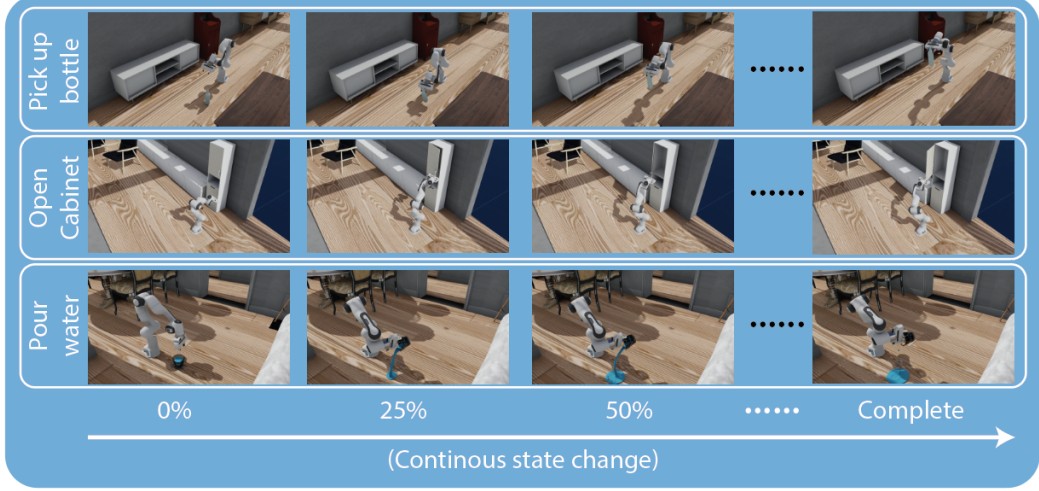

Figure 10: Continous control and goal state

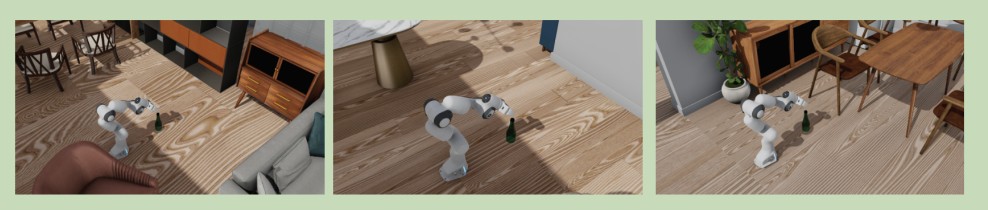

Figure 11: Scene variations

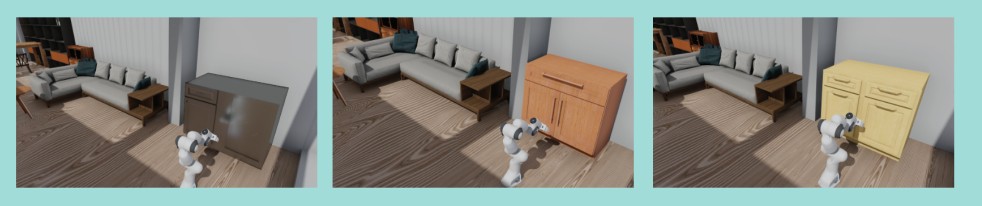

Figure 12: Object variations

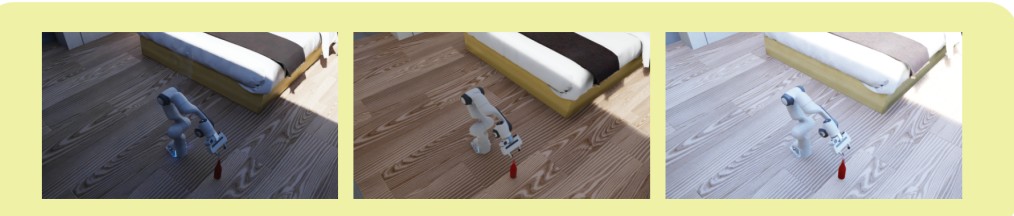

Figure 13: Lighting variations

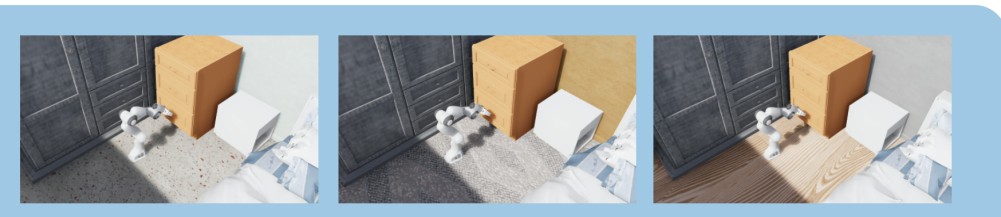

Figure 14: Material variations

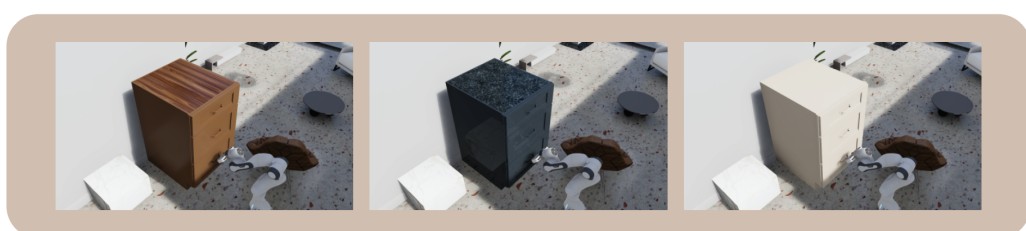

Figure 15: Object variations

