# OpenReview forum: "ARNOLD: A Benchmark for Language-Grounded Task Learning with Continuous States in Realistic Scenes"
_robot-learning.org/CoRL/2022/Workshop/LangRob — LangRob 2022 Spotlight_

### Official Review · Reviewer_Kimy · 2022-11-12
**Good benchmark**

**Rating:** 8
**Confidence:** 4

**Review:**

This paper presents ARNOLD, a language-grounding benchmark for achieving continuous state goals. The benchmark includes 8 tasks in Issac Gym, where each task involves manipulating an object to a desired continuous state like “opening the door to 25%” or “rotate the block to 45 degrees”.

Strengths:
+ ARNOLD addresses as a good gap in vision-language manipulation, i.e. achieving continuous goals.
+ The benchmark is set in a photo-realistic environment, and upon release, it should be of interest to the robotics+language community.
+ Table 1 does a good job of comparing against prior benchmarks.
+ Limitations are clearly acknowledged.

Weaknesses and Suggestions:
- Baselines 6D CLIPort and PerAct are hinted at, but it would be great to get some quantitative results with them or other end-to-end approaches.
- Any specific motivation behind discretizing the goal space into 25%, 50%, 75%, and 100% increments? Why not pick an arbitrary continuous value and evaluate with a given error tolerance?
- 8 tasks are already great, but it would be awesome to have a few more, especially multi-object scenes. Diverse scenes will help improve the diversity of language instructions.

---

### Official Review · Reviewer_NPuW · 2022-11-12
**Blind Review#1**

**Rating:** 8
**Confidence:** 4

**Review:**

Summary: The paper presents a novel benchmark for language grounding with photo-realistic scenes and continuous robot actions and object states. Given the contribution of building such a benchmark, I would recommend accepting it, and the future path of this work is as well relatively clear and promising to me.

Pros:

- The benchmark provides abundant manipulation data and has advantages over each existing benchmark in different aspects, which fills the gap in language-conditioned task learning and makes it a valuable benchmark.
- A comparison is conducted over most language-conditioned task learning benchmarks, which indicates the adequate amount of effort in background research.
- The proposed data collection process is logically sound and explained in detail with supplementary material in the appendix.
- The contributions of the paper are stated clearly, and the limitations are discussed in depth.
- The language this paper used is easy to follow.

Cons:

- There lacks a baseline model to provide baseline performance for this benchmark, and it is hard to validate the proposed work and estimate the difficulty of the task, such as the feasibility of the dataset for learning continuous state language grounding tasks.
- Table 2 suggests that four goal states are specified for each task, and my concern is the intervals are probably defined too large to claim the object state is continuous and fine-grained. For example, Figure 10 shows the joint angles vary non-trivially between different goal states of the same task. Could it make the different goal states essentially different tasks?

Other comments:

- Nature language has errors, but it should still be meaningful to collect them. I would suggest removing the statement “at the same time…significant errors” in the limitations section. It will be generally good to see natural language instructions in future work or at least introduce manual noise through approaches such as monolingual translation.
- I would suggest providing more folds with different combinations of variables in 3.2.4 for conducting a solid and insightful evaluation, e.g., one fold for evaluation in the all-seen environment, one fold for evaluation in the all-novel environment, and other folds for combinations of novel or seen textures, objects, and language.
- The 2-second limit described in the evaluation metrics seems to be arbitrary.
- Having continuous states for both rigid and fluid in simulation is a great add-on and worth emphasizing.
- Before releasing the challenge, I would like to suggest the author(s) to compare the performance of the baseline model on ARNOLD to other discretized and non-photo-realistic datasets to further strengthen the conclusion in the future.

Writing:

- Section 3 is too long and better to be divided into multiple sections.
- There is no future work in the conclusion&future work section but in the limitations section.
- More statistics of the collected dataset could be presented in the body of the paper, such as the statistics of the dataset in the appendix. Also, the instruction generation part in the appendix can be added to the body as well.
- Section 4 discussed two existing works but did not reproduce both, which should rather be considered as part of the related work.
- The current benchmark does not contain navigation tasks, and it should not be framed as for general robot language grounding at this moment.

---

### Decision · Program_Chairs · 2022-11-15

Accept (Spotlight)